# Tilorone-Dihydrochloride Protects against Rift Valley Fever Virus Infection and Disease in the Mouse Model

**DOI:** 10.3390/microorganisms10010092

**Published:** 2021-12-31

**Authors:** Kendra N. Johnson, Birte Kalveram, Jennifer K. Smith, Lihong Zhang, Terry Juelich, Colm Atkins, Tetsuro Ikegami, Alexander N. Freiberg

**Affiliations:** 1Department of Microbiology and Immunology, University of Texas Medical Branch, Galveston, TX 77555, USA; kennjohn@utmb.edu; 2Department of Pathology, University of Texas Medical Branch, Galveston, TX 77555, USA; bkkalver@utmb.edu (B.K.); jeksmith@utmb.edu (J.K.S.); lihzhang@utmb.edu (L.Z.); tljuelic@utmb.edu (T.J.); cpa53@dls.rutgers.edu (C.A.); teikegam@utmb.edu (T.I.); 3Center for Biodefense and Emerging Infectious Diseases, University of Texas Medical Branch, Galveston, TX 77555, USA; 4Institute for Human Infections and Immunity, University of Texas Medical Branch, Galveston, TX 77555, USA; 5Sealy Institute for Vaccine Sciences, University of Texas Medical Branch, Galveston, TX 77555, USA

**Keywords:** Tilorone-dihydrochloride, Rift Valley fever virus, Bunyavirales, *Phenuiviridae*, Viral Hemorrhagic fever, broad-spectrum antiviral

## Abstract

Rift Valley fever (RVF) is a mosquito-borne zoonotic disease endemic to Africa and the Middle East that can affect humans and ruminant livestock. Currently, there are no approved vaccines or therapeutics for the treatment of severe RVF disease in humans. Tilorone-dihydrochloride (Tilorone) is a broad-spectrum antiviral candidate that has previously shown efficacy against a wide range of DNA and RNA viruses, and which is clinically utilized for the treatment of respiratory infections in Russia and other Eastern European countries. Here, we evaluated the antiviral activity of Tilorone against Rift Valley fever virus (RVFV). In vitro, Tilorone inhibited both vaccine (MP-12) and virulent (ZH501) strains of RVFV at low micromolar concentrations. In the mouse model, treatment with Tilorone significantly improved survival outcomes in BALB/c mice challenged with a lethal dose of RVFV ZH501. Treatment with 30 mg/kg/day resulted in 80% survival when administered immediately after infection. In post-exposure prophylaxis, Tilorone resulted in 30% survival at one day after infection when administered at 45 mg/kg/day. These findings demonstrate that Tilorone has potent antiviral efficacy against RVFV infection in vitro and in vivo and supports further development of Tilorone as a potential antiviral therapeutic for treatment of RVFV infection.

## 1. Introduction

Rift Valley fever virus (RVFV) is a mosquito-borne RNA virus of the *Phenuiviridae* family (order Bunyavirales) endemic to Sub-Saharan Africa and is the causative agent of Rift Valley fever (RVF) [1,2]. RVF is most commonly a disease of domestic ruminants such as cattle, goats, and sheep, primarily presenting as an acute fever [3]. In particularly susceptible species, such as sheep, it causes fetal malformation and abortion storm in pregnant animals and has high newborn mortality rates up to 100% [3]. Along with regular outbreaks in livestock, RVF causes spillover events into human populations [4,5,6,7,8]. Most human cases of RVF present as a self-limiting febrile illness. However, a small percentage of patients progress to more severe manifestations such as hemorrhagic fever, retinitis or encephalitis [9,10,11,12]. The emergence of RVFV into new regions such as Egypt, the Arabian Peninsula, Madagascar, and the Comoros has brought attention to the risk for even wider spread into non-endemic areas [13]. As RVFV has the potential to cause serious agricultural and health problems, it has been classified as a Category A Priority Pathogen by the National Institutes of Allergy and Infectious Diseases in the United States, a Blueprint Priority Disease by the World Health Organization, as well as an overlap select agent by the Centers for Disease Control & Prevention and United States Department of Agriculture [14,15].

Despite the severe outcomes of past and current RVF outbreaks, no FDA-approved vaccines or therapeutics for humans are available and the current treatment strategy is mainly supportive care. The broad-spectrum antiviral Ribavirin was proposed to be of benefit in severe RVF cases; however, an efficacy trial undertaken during the 2000 outbreak in Saudi Arabia was terminated due to adverse effects [16]. In addition, results from studies in animal models suggest that treatment with Ribavirin increases the risk of delayed-onset neurologic complications [16]. Although many robust animal models are available for RVFV, only a limited number of antivirals have been evaluated in vivo thus far and demonstrated varying efficacy [17]. 

Tilorone-dihydrochloride (Tilorone, 2,7-Bis[2-(diethylamino)ethoxy]-9H-fluoren-9-one) is a broad-spectrum antiviral that was identified over 50 years ago, and is used clinically in Russia, Ukraine, Kazakhstan, Belarus, Armenia, Georgia, Kyrgyzstan, Moldova, Turkmenistan, and Uzbekistan [18]. Tilorone is utilized to treat a variety of viral disease indications, such as influenza, acute respiratory viral infection, viral hepatitis, and viral encephalitis and is included on a list of essential medicines of the Russian Federation [19,20,21,22,23]. In addition to its clinical application, Tilorone has demonstrated in vitro and in vivo efficacy against a wide range of viral families, such as filoviruses, flaviviruses, coronaviruses, and alphaviruses [18,24,25,26,27]. Most recently, it demonstrated in vitro activity against SARS-CoV-2 in the nanomolar range [27]. Tilorone was initially thought to act through activation of innate immunity signaling pathways, particularly those involved in production of interferon (IFN) [19,28], although recent investigations have suggested its potential mechanism of action to include binding to viral surface glycoproteins and lysosomotropism [18,27,29]. In this study, we assessed the ability of Tilorone to inhibit RVFV infection, and could demonstrate that it interferes with RVFV in the low micromolar range and has protective efficacy in a lethal mouse model.

## 2. Materials and Methods

### 2.1. Cells and Viruses

Vero CCL81 cells and A549 cells were purchased from the American Type Culture Collection (ATCC, Manassas, VA, USA). Vero CCL81 cells were grown and maintained in minimal essential media (MEM) and A549 cells were maintained in Dulbecco’s minimal essential media (DMEM), both supplemented with 10% Fetal Bovine Serum (FBS) at 37 °C under 5% CO_2_. During viral infections, FBS concentration was lowered to 2% and 1% penicillin/streptomycin (Corning) was added.

Pathogenic RVFV wild-type strain ZH501 and the original live-attenuated vaccine strain MP-12 were used in in vitro and in vivo studies, respectively (obtained from Drs C.J. Peters and John Morrill, University of Texas Medical Branch, UTMB). Viral titers were determined by plaque assay. Briefly, Vero CCL81 cells were infected with serial 10-fold dilutions of virus containing samples for one hour and then overlaid with tragacanth (0.8%)/MEM, supplemented with 2% FBS and 1% penicillin/streptomycin. After three days, the overlay was removed and cells were stained with 0.2% crystal violet diluted in 10% neutral buffered formalin for at least 20 min at room temperature. Plates were washed with water, dried, plaques enumerated and viral titers reported as plaque forming units per mL (PFU/mL). All work with infectious RVFV ZH501 virus was conducted in the Robert E. Shope or Galveston National Laboratory biosafety level 4 (BSL-4) laboratories at the UTMB.

### 2.2. Compounds

Tilorone dihydrochloride was obtained from Sigma-Aldrich (St. Louis, MO, USA). For in vitro experiments, Tilorone was diluted in cell culture media. For in vivo experiments, Tilorone dilutions were prepared in 20% Solutol (Kolliphor HS 15).

### 2.3. In Vitro Virus Yield Reduction Assay

Vero CCL81 or A549 cells were infected with RVFV MP-12 or RVFV ZH501 at a multiplicity of infection (MOI) of 0.1 for one hour at 37 °C and 5% CO_2_ with frequent rocking every 15 min. Virus was then removed, cells washed with DPBS and overlaid with fresh media supplemented with 2% FBS and 1% penicillin/streptomycin and half-log dilutions of Tilorone (100 to 0.032 μM). Cell culture supernatant aliquots were collected at 24 h post-infection (HPI) and titrated. Virus yield reduction was calculated as percent reduction of viral titers compared to untreated controls. Cellular cytotoxicity of Tilorone was determined in the absence of viral infection using a neutral red based in vitro toxicology assay kit (Sigma-Aldrich; St. Louis, MO, USA). All experiments were performed in triplicate wells.

The 50% effective concentration (EC_50_) was determined as the concentration at which viral titers were 50% of the untreated controls at the respective time point and the 50% cell cytotoxic dose (CC_50_) was the Tilorone concentration leading to 50% cell cytotoxicity. Both values were calculated using regression analysis (Graphpad Prism V8). The selectivity index (SI) was calculated using the formula SI = CC_50_/EC_50_.

### 2.4. In Vitro Time of Addition Experiments

Vero CCL81 or A549 cells were infected with RVFV MP-12 or RVFV ZH501 at an MOI of 0.1 as described above. At 1 HPI, inoculum was removed, cells washed with DPBS, and fresh media added. Tilorone was added at −1, 0, 1, 6, and 12 HPI and supernatant was sampled at 24 HPI. Viral titers were determined via plaque assay. All experiments were performed in biological triplicates.

### 2.5. Mouse Efficacy Studies

Six-to-eight week-old female BALB/c mice (Envigo) were utilized for antiviral efficacy studies. Mice were challenged with 100 PFU of RVFV ZH501 by intraperitoneal (IP) injection and the viral dose verified by standard plaque assay. In the first experiment, animals (n = 10 per group) received either 30 mg/kg/day or 60 mg/kg/day of Tilorone in 100 μL via IP injection, with treatment continuing daily for 9 days after initiation. Dosing began either 24 h before or immediately after infection on the day of challenge. In the second experiment, animals (n = 10 per group) were dosed once daily with 45 mg/kg/day for a 9-day period, with dosing beginning immediately after infection, 1 day post-infection (DPI), 2 DPI, or 3 DPI. In each experiment, a virus only control group (n = 10) received the solutol vehicle solution for a 9-day dosing period, beginning immediately after infection.

In all studies, animals were monitored daily for clinical signs of disease for 21 DPI. Once animals reached a moribund state, they were euthanized. Body weights were taken daily for the first 10 days and then every 3rd day until end of study. In both studies, terminal bleeds were collected from moribund animals to determine viremia by plaque assay. Tissues (brain and liver) were collected from survivors at termination of the study and homogenized in TRIzol (Life Technologies, Carlsbad, CA, USA) to evaluate for presence of RVFV genome. For the second experiment, tissues (brain, liver) from vehicle-treated animals and survivors were collected and one half of the tissues either processed in TRIzol (Life Technologies) or fixed in 10% Neutral Buffered Formalin for histopathological analysis. For each manipulation (viral infection or drug administration), animals were anesthetized with isoflurane (Piramal, Mumbai, India).

### 2.6. Animal Ethics Statement

All procedures were conducted under animal protocols approved by the UTMB Institutional Animal Care and Use Committee and complied with USDA guidelines in an AAALAC-accredited lab. Animals were housed in microisolator caging equipped with HEPA filters in the BSL-4 laboratories at UTMB.

### 2.7. qRT-PCR

qRT-PCR was utilized to qualitatively evaluate presence of viral RNA in tissue samples. RNA was extracted from tissues homogenized in TRIzol reagent using Direct-zol RNA Miniprep kits (Zymo Research, Orange, CA, USA). qRT-PCR assays were run using QuantiFast RT-PCR mix (Qiagen, Hilden, Germany) using primer and probes targeting the RVFV L gene (TIB MOLBIOL). qRT-PCR was performed using the following cycle: 10 min at 50 °C, 5 min at 95 °C, and 40 cycles of 10 s at 95 °C and 30 s at 60 °C using a BioRad CFX96 real time system.

### 2.8. Histopathological Analysis

Formalin fixed tissues were embedded in paraffin at the UTMB Research Histopathology Core. Embedded tissues were sectioned and stained with haematoxylin and eosin (H&E). Images were obtained using an Evos XL Core microscope (Life Technologies).

### 2.9. Statistical Analysis

All statistical analysis was completed using Prism (GraphPad Software, San Diego, CA, USA). Dose–response curves were developed using nonlinear regression. Comparisons of viral titers in time of addition assays were subjected to a two-way repeated measure analysis of variance (ANOVA) with a Tukey post-test. Survival curves were compared using the Mantel-Cox log-rank test. Serum titers were compared using a one-way ANOVA.

## 3. Results

### 3.1. Tilorone Inhibits Rift Valley Fever Virus Replication In Vitro

To determine the potential antiviral efficacy of Tilorone for RVFV, we employed virus yield reduction assays. Initial assays were performed using the vaccine strain RVFV MP-12 in both Vero CCL81 and A549 cells, respectively. Due to the previous reports that Tilorone might act through activation of IFN-related innate immunity signaling pathways [19,28], we chose to evaluate its antiviral activity in both, type-I IFN-deficient Vero CCL81 and type-I IFN-competent A549 cells. Cytotoxicity at the highest concentration tested was minimal at 24 HPI for both cell types with a CC_50_ > 100 μM (Table 1). With increasing incubation times for 48 and 72 HPI, cytotoxicity increased and was higher in A549 cells compared to Vero CCL81 cells. Next, the effects on RVFV replication were determined at 24 HPI after addition of Tilorone. For both cell lines, Tilorone treatment resulted in reduced viral titers in a dose-dependent manner (Figure 1). Analysis of the dose–response curves resulted in EC_50_ values for RVFV MP-12 of 0.67 μM in Vero CCL81 cells and 1.41 μM in A549 cells (Figure 1A,B). The Selective Index (SI) values were >149 in Vero CCL81 and >71 in A549 cells, respectively (Table 2). We then confirmed the activity of Tilorone against the pathogenic RVFV ZH501 strain (Figure 1C,D). Here, the EC_50_ values were 6.45 μM in Vero CCL81 and 6.31 μM in A549 cells, respectively, with SI values of >16 in both Vero CCL81 and A549 cells (Table 2). These data demonstrate that RVFV is sensitive to treatment with Tilorone, with EC_50_’s that are consistent with those described for other viruses [18].

### 3.2. Delayed Treatment Efficacy of Tilorone on Rift Valley Fever Virus Infection In Vitro

Next, the inhibitory effect of Tilorone on RVFV replication was determined in a post-exposure treatment scenario. Vero CCL81 or A549 cells were infected with RVFV MP-12 and Tilorone at 50 μM added at varying times of infection (−1 to 12 HPI). This concentration was chosen, because a near 100% inhibition was achieved in the dose–response curves (Figure 1A,B). Virus titers were then determined at 24 HPI (Figure 2). In RVFV MP-12-infected A549 cells, viral titers at 24 HPI were close to or below the limit of detection of the plaque assay when Tilorone was added up to 6 HPI. In contrast, untreated cells displayed a titer of ~3.5 × 10^4^ PFU/mL. Addition of Tilorone at 12 HPI resulted in significantly reduced titers below 10^3^ PFU/mL (Figure 2A). Similar trends were observed at 24 HPI in Vero CCL81 cells. The delayed treatment antiviral activity of Tilorone on RVFV replication was then confirmed for the pathogenic strain ZH501 (Figure 2B). In A549 cells, initiation of treatment with 50 μM up to 6 HPI reduced viral titers to around 10^3^ PFU/mL at 24 HPI, compared to ~3 × 10^6^ PFU/mL in untreated cells (Figure 2B). As observed with RVFV MP-12, treatment at 12 HPI still significantly reduced RVFV ZH501 titers compared to untreated cells, although to a lesser extent than earlier treatment. In Vero CCL81 cells, a time-dependent antiviral effect could be observed with 50 μM (Figure 2B). Pre-treatment reduced RVFV ZH501 titers to approximately 10^4^ PFU/mL compared to 10^7^ PFU/mL in untreated cells. While this reduction in titer became less pronounced for each subsequent time of addition, even addition at 12 HPI significantly reduced viral titers by at least one log. Overall, these results demonstrate that while Tilorone is most effective at inhibiting RVFV replication when added within 6 HPI, delaying treatment in vitro for up to 12 HPI still leads to a significant reduction in viral load.

### 3.3. Administration of Tilorone Reduces RVFV-Induced Mortality in the BALB/c Mouse Model

Encouraged by the observed ability of Tilorone to inhibit RVFV replication, we evaluated the antiviral efficacy in the BALB/c mouse model for RVFV ZH501. The maximum tolerated dose (MTD) in BALB/c mice was previously investigated by Ekins et al. and found to be 100 mg/kg of body weight in a single dose IP injection [24]. The MTD study found that doses of 10, 50 and 100 mg/kg were associated with 100% survival although adverse events including ruffled fur and hunched posture were noted even at the lower dosages. In this study, it was demonstrated that once daily dosing of 50 mg/kg/day for 8 consecutive days resulted in 90% survival in Ebola virus (EBOV)-infected mice [24]. Based on these findings, we decided to evaluate Tilorone at a high (60 mg/kg/day) and a low (30 mg/kg/day) dose. Groups of 10 BALB/c mice (female, 6–8 weeks old) were infected with RVFV ZH501 via the IP route with 100 PFU. Treatment with Tilorone was performed via the IP route, once daily for a total of 9 doses and was initiated either at 24 h prior to infection (−24 HPI pre-treatment), or immediately after infection (0 HPI co-treatment). A virus control group received vehicle solution only. Figure 3A shows that infected animals receiving vehicle solution uniformly succumbed to disease by 9 DPI. The two groups given 30 mg/kg/day showed 80% survival when given immediately after infection and 40% survival when given 24 h prior to infection (Figure 3A). It should be noted that one mouse in the 30 mg/kg/day in the −24 HPI pre-treatment group did not recover from anesthesia after the pre-treatment dosing was administered, reducing the group size to a total of 9 animals. In the 60 mg/kg/day groups, 40% of mice dosed at 24 h prior to infection survived, while no survivors were detected in the 0 HPI co-treatment group (Figure 3A). This unexpected result is presumably due to drug toxicity from a non-optimal dosing regimen and might require further investigation. Clinical adverse observation in the 30 mg/kg/day groups was minimal and late-onset, consisting primarily of scruffy coat and lethargy, and began only 9 days post infection for the co-treatment group and 11 DPI for the pre-treatment group. Non-survivors from the 30 mg/kg/day groups maintained low clinical scores until immediately before developing severe disease. Severe disease symptoms were primarily neurological symptoms characteristic of late-onset RVFV disease, such as rear leg paralysis and ataxia. On the other hand, the 60 mg/kg/day groups displayed some signs of drug toxicity from the repeated dosing. Here, pre-treatment with 60 mg/kg/day led to observation of scruffy coat even before inoculation with RVFV, a clear indication of drug-mediated toxicity. For both 60 mg/kg/day groups, the scruffy coat was maintained throughout the entire 9-day course of Tilorone treatment, regardless of survival outcome. More severe symptoms began as early as 2 days and included hunching and lethargy, similar to vehicle treated mice. On days 8 and 9 PI, a number of clinical symptoms were observed, including those characteristic of RVFV infection (e.g., irregular breathing, lethargy, and orbital tightening), as well as those not typically associated with RVFV infection (e.g., diarrhea and distended abdomen). During necropsy of non-survivors, it was noted that the distended abdomen was filled with a cloudy liquid and intestine was enlarged and pale. These abdominal symptoms could result from drug-related toxicity. Control mice receiving vehicle solution and mice receiving 60 mg/kg/day Tilorone immediately after infection rapidly started losing weight, with minimal weight loss in all other groups, supporting the hypothesis of toxicity caused by non-optimal dosing (Figure 3B). Serum samples were collected at the time of termination from moribund vehicle-treated control mice, as well as from non-survivors from each treatment group, and viremia was evaluated by plaque assay (Figure 3C). Average viremia levels in the vehicle-treated animals were 1.4 × 10^6^ PFU/mL. In contrast, no viremia was detected in any of the collected terminal samples from the three treatment groups that demonstrated significant increases in survival, even though these samples were from the non-survivors in their respective groups. On the other hand, in the 60 mg/kg/day 0 HPI co-treatment group, 4 out of the 5 animals had viremia between 2.5 × 10^2^ and 1.7 × 10^4^ PFU/mL (Figure 3C). Additionally, the brain and liver of all 8 survivors in the 30 mg/kg/day dosing group (0 HPI co-treatment group) were evaluated for the presence of viral genome using a qualitative positive/negative RT-PCR assay and no viral RNA could be detected (data not shown). These findings indicate that Tilorone is capable of suppressing viral replication and promoting clearance of RVFV in the mouse model.

### 3.4. Tilorone Has Limited Window of Therapeutic Efficacy against Lethal RVFV Infection in BALB/c Mice

Due to the observed toxic side effects with the 60 mg/kg/day groups, we performed a follow-up study in which the antiviral efficacy of Tilorone was evaluated at 45 mg/kg. In addition, we also sought to determine the therapeutic window of efficacy of Tilorone. As in the first study, groups of 10 BALB/c mice were infected with RVFV ZH501 via the IP route with 100 PFU. Dosing with Tilorone was performed via the IP route, once daily for a total of 9 dosings, and treatment with 45 mg/kg/day was initiated at 0, 1, 2, or 3 DPI. As expected, all non-protected vehicle-treated animals succumbed to disease by 9 DPI (Figure 4A). Similar to the previously described experiment, treatment administered immediately after infection resulted in 70% survival (Figure 4A). At this dosing concentration, no toxicity was observed compared to the previously tested 60 mg/kg groups. Delayed administration showed a reduced protective efficacy, with 30% protection and delayed onset of disease when dosing was initiated at 1 DPI. In the 2 DPI group, survival was reduced to 10%, and no protection was observed at 3 DPI. Non-protected mice and mice receiving Tilorone treatment 2 or 3 DPI rapidly started losing weight, while weight loss in all other groups was reduced (Figure 4B). In the 0 and 1 DPI treatment groups, no viremia was detected in terminal blood from moribund animals. Contrastingly, viremia was detected in several blood samples from the 2 and 3 DPI treatment groups (Figure 4C). Tissues from moribund animals of the vehicle group, as well as from survivors were collected in TRIzol and evaluated for the presence of viral genome. Like in the previous study, no viral RNA was detected in any survivor, while viral RNA was detectable in all vehicle-treated mice (data not shown).

### 3.5. Histopathological Evaluation

To determine pathological changes in Tilorone-treated mice, liver and brain tissues from the 45 mg/kg treatment study were collected from moribund animals and survivors and H&E staining performed (Figure 5). In vehicle-treated animals, pathological changes were found in livers. One mouse showed enlarged ballooned hepatocytes throughout the lobule with moderate infiltration of mononuclear and polymorphonuclear cells in the sinusoid, which is likely associated with hepatocellular regeneration after the stage of acute liver injury (Figure 5A). Two other mice in the vehicle-treated group showed necrotic or apoptotic changes, characterized by pyknosis, karyorrhexis, or karyolysis of hepatocytes in throughout hepatic lobes, whereas little infiltration of inflammatory cells was found in necrotic lesions. (Figure 5B). In Tilorone-treated survivors, indistinguishable histology in liver sections between treatment groups were observed. Although most parts of liver sections were normal (Figure 5C), extramedullary hematopoiesis such as erythroid cells or megakaryocytes were occasionally found (Figure 5D,E). Although rare, mild infiltration of neutrophils was also observed (Figure 5F).

## 4. Discussion

Rift Valley Fever is primarily a severe disease of ruminant livestock but can also cause moderate to severe illness in humans. Since the first report of the disease among livestock in Kenya in 1915, RVFV has expanded its range outside of Africa to include Saudi Arabia, Yemen, Madagascar, and the Comoros [10,30,31,32,33]. Additionally, the range of the mosquitoes capable of carrying and transmitting the virus now includes the Americas and Europe [15,34,35]. While most human cases are mild and self-limiting and do not require intensive care, there are currently no therapeutics available for treatment of the more severe manifestations. Due to its potential to cause significant economic losses in the livestock industry and morbidity in humans, RVFV is categorized as an HHS/ USDA select agent and requires higher biocontainment facilities (biosafety level 3 enhanced or 4). The restrictions in handling infectious RVFV might have been a contributing factor in the dearth of antiviral therapeutics that have been evaluated until now. Some of these that have shown efficacy in animal models include the nucleoside analog Ribavirin and nucleotide analog Favipiravir [36,37]. Ribavirin demonstrated partial protection (70% survival) after subcutaneous infection in the murine model, although not after aerosol infection in the same model [37]. On the other hand, Favipiravir treatment led to 92% survival after aerosol infection in the Wistar-Furth rat model [36,38].

While not FDA approved, Tilorone-Dihydrochloride is approved and regularly used for treatment of influenza in several eastern European countries, indicating it is safe for use in humans [18,21]. While Tilorone has never been evaluated for safety and efficacy to meet current FDA standards, it should be noted that Tilorone analogs have undergone numerous clinical trials in Russia and have a track record of safe usage in humans outside the US [19,20,21,22,23]. A machine learning computational screen identified Tilorone as a potential inhibitor of EBOV, and it was found to be 90–100% effective in the mouse model, depending on the dosing conditions [24]. This promising result renewed interest in Tilorone in the US and it has been investigated for activity against a variety of viruses, including SARS-CoV-2 [27].

In this present study, we sought to evaluate the antiviral efficacy of Tilorone, in both in vitro and in vivo models, for RVFV. In vitro, Tilorone displayed antiviral activity in the low micromolar range. For the vaccine strain RVFV MP-12, the EC_50_ were 0.67 μM and 1.41 μM for Vero CCL81 and A549 cells, respectively, while higher concentrations were required for inhibition of the wildtype strain ZH501 with 6.45 μM and 6.31 μM, respectively. Previously, screening by another group showed little to no efficacy of Tilorone against RVFV ZH501 in Vero 76 cells [39]. This might have been due to different assays. At first glance, these differences in antiviral efficacy between IFN-competent A549 cells and IFN-deficient Vero cells against wildtype RVFV ZH501 may appear to be related to the capacity for IFN induction. Tilorone was indeed initially identified in 1970 as an IFN inducer and a number of early studies indicated this activity as a potential mechanism of action [19,28,40]. However, the fact that antiviral activity can still be observed in Vero CCL81 cells indicates that Tilorone might function through an additional mechanism of action, in addition to IFN induction. Indeed, recent studies revealed that varying antiviral efficacy of Tilorone between cell types is consistent with its activity against other viruses and that observed differences in efficacy do not always correlate with IFN competency. For example, Tilorone is effective against EBOV in HeLa cells, which are IFN-competent, but not in Vero 76 cells [18]. For SARS-2 coronavirus (SARS-CoV-2), antiviral activity was demonstrated in A549-ACE2 cells and Vero 76 cells, and to a lesser extent in Caco-2 and Calu-3 cells, but interestingly not in Vero E6 cells [27]. Tilorone also inhibited replication of MERS-CoV and Chikungunya virus in Vero 76 cells, although other cell lines were not tested for these viruses [41]. Due to the fact that Vero cells are IFN-deficient, Tilorone might display additional mechanisms of viral inhibition contributing to widely differing efficacies of Tilorone against different viruses across different Vero lineages. Ultimately, for wildtype RVFV ZH501, the slightly higher EC_50_ values determined in IFN-competent A549 cells over IFN-deficient Vero CCL81 cells suggests that a partial role for innate immune activation could be a contributing factor as a mechanism of action.

Lysosomotropism has been described as a mechanism of action for broad-spectrum antivirals [42]. Tilorone is an amphiphilic cationic compound with demonstrated lysosomotropic activity, which may be an important factor in its mechanism of action [39]. It has been found to increase lysosomal pH and inhibit the ATP-dependent acidification of lysosomes in fibroblasts, which are important characteristics of lysosomotropic compounds [43]. Biochemical or metabolic differences across different cell lines could explain altered responses to Tilorone’s lysosomotropism and contribute to differences in antiviral activity across cell types. Another proposed mechanism of action is direct antiviral activity through receptor binding. Tilorone has been experimentally verified to strongly bind the EBOV glycoprotein, potentially helping block viral entry [29]. It also binds the SARS-CoV-2 spikeprotein receptor-binding domain (RBD); however, neutralization was less than 50% in a VSV pseudotype assay, suggesting that RBP binding may not be important for Tilorone’s antiviral activity against SARS-CoV-2 [27]. It seems likely that some combination of these mechanisms acts synergistically to result in antiviral activity, but future studies are required to further elucidate the exact mechanisms of action by which Tilorone interferes with RVFV infection.

In vitro time of addition assays demonstrated that delayed treatment up to 6 h can significantly reduce RVFV replication to levels near the limit of detection of the plaque assay protocol. Delaying treatment by 12 h still significantly reduced virus titers compared to untreated controls. These results indicate that Tilorone might be used in both, prophylactic and therapeutic approaches. One observation that was noted, is that Tilorone caused cytotoxic effects in both cell types by 48 and 72 h after treatment (Table 1). Further evaluation on the potential effect of Tilorone-induced cytotoxicity on viral replication needs to be evaluated in future experiments.

Due to the requirement for high containment facilities for use of pathogenic RVFV ZH501, a limited number of antiviral compounds have been evaluated in vivo [17]. In this study, Tilorone was found to provide 80% protection and delaying time to death in the lethal BALB/c mouse model of RVFV ZH501 infection. The first study examined doses of 30 and 60 mg/kg/day, doses which were examined and found efficacious to some extent in the mouse model against EBOV [24]. Here, a 30 mg/kg dose of Tilorone given daily and beginning either 2 or 24 h after EBOV infection led to 100% survival in both groups. Similar trends were observed in our data, in which the greatest survival was seen after a 30 mg/kg/day dose of Tilorone initiated immediately after infection although it is still unclear why pre-treatment with 30 mg/kg/day resulted in only 40% survival, compared to 80% survival when administered immediately after infection. For the 60 mg/kg/day groups, increased survival was observed after initiation of treatment 24 h before infection, rather than concurrent with infection. As in the case of the EBOV study, signs of drug toxicity from repeated dosing in the 60 mg/kg/day dose groups was found in the present study. It was postulated that results were due to drug toxicity from a nonoptimal dosing regimen leading to drug accumulation and may not completely reflect a lack of efficacy [24]. We adjusted dosing to 45 mg/kg/day to investigate the window of efficacy, and results indicated that initiation of treatment within 24 h after infection lead to some level of efficacy, whereas survival was reduced to 10% the later treatment was initiated. No significant difference in viremia between the 2 and 3 DPI Tilorone and vehicle groups was found, suggesting lack of viral inhibition in these treatment groups. Additionally, no viral RNA was detected in tissues for any survivors and no replicating virus was detected in serum samples from groups with significant survival across both in vivo studies. While there were also some non-survivors in both untreated and treated groups in which viremia was not detected, this is not unexpected due to the biphasic nature of RVFV infection in mice. In early stages of infection, viral titers in blood and liver are typically high [44]. However, in those mice that survive the initial phase of infection, virus is cleared from the blood and infection is characterized by neuroinvasion [44]. Our observations in non-survivors of all groups that succumbed later in disease mirror this. For example, the non-survivors for which no viremia was detected in sera were those that succumbed later in infection and those for which viremia was detected, succumbed earlier. In future studies, it will be valuable to collect sera at timepoints throughout the course of infection, particularly early in infection, to analyze the effects of Tilorone treatment on viral titers at peak viremia.

Due to potential drug toxicity and tolerability concerns, optimization of the dosing regimen will require further investigation. However, up to 80% survival are encouraging and demonstrate equal or improved efficacy compared to other antivirals evaluated in the RVFV mouse model [38,45]. Treatment with 200 μM Favipiravir (or T-705) led to 80% survival in BALB/c mice with ≤40% survival for all other conditions tested, including combination treatment with Ribavirin [38]. Optimization of dosing conditions of Ribavirin provided up to 80% survival in Swiss Webster mice [46]. Pretreatment with BCX4430 led to up to 60% survival in the C57/BL6 model and Rapamycin treatment provided up to 50% survival in BALB/c mice [45,47]. Differences in mouse species and routes of infection and dosing conditions between these studies make it somewhat difficult to directly compare antiviral efficacy, but overall, Tilorone could be further considered for future development as a potential therapeutic for treatment of RVFV. Drug toleration issues in mice, with administration through the IP route may also not be reflective of those in humans. In counties where Tilorone derivatives are utilized as therapeutics in patients, administration is performed through the oral route, which, in addition to species differences, could contribute to better toleration [18].

## 5. Conclusions

Altogether, the results presented here demonstrate the in vitro and in vivo efficacy of Tilorone-Dihydrochloride against RVFV infection. Our data are providing a foundation for additional studies regarding the optimization of doses, routes of administration, as well as timing of treatment after infection. In addition, combinatorial treatment with antiviral compounds that display a different mechanism of action and that have shown efficacy against RVFV, should be considered.

## Figures and Tables

**Figure 1 microorganisms-10-00092-f001:**
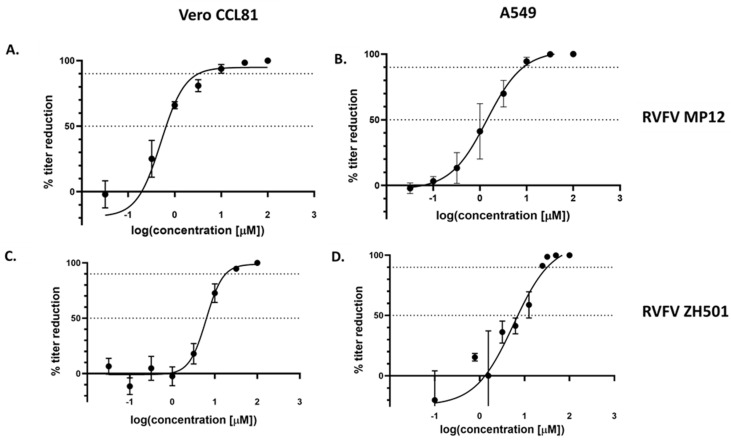
In vitro dose-response of Tilorone against RVFV. Vero CCL81 cells (**A**,**C**) or A549 cells (**B**,**D**) were infected with RVFV MP12 (**A**,**B**) or RVFV ZH501 (**C**,**D**) at an MOI of 0.1 for 1 h. Cell culture media with serial 10-fold dilutions of Tilorone was added at 1 h post infection (HPI). Reduction in virus yield was determined by plaque assay in cell culture supernatant collected at 24 HPI. Data is representative of two individual experiments, each with three biological replicates.

**Figure 2 microorganisms-10-00092-f002:**
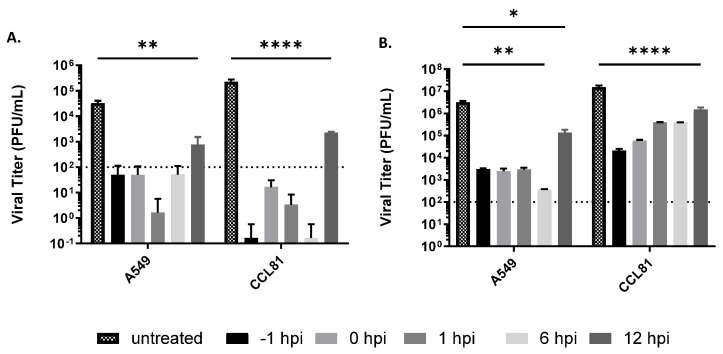
Delayed treatment in vitro efficacy of Tilorone against RVFV infection. A549 cells or Vero CCL81 cells were infected with RVFV MP12 (**A**) or ZH501 (**B**) at an MOI of 0.1 and treated with 50 μM of Tilorone at the time points indicated. Virus titer in cell culture supernatant was evaluated by plaque assay at 24 h after infection. The dotted line represents the limit of detection in the plaque assay. * *p* < 0.05, ** *p* < 0.01, **** *p* < 0.0001.

**Figure 3 microorganisms-10-00092-f003:**
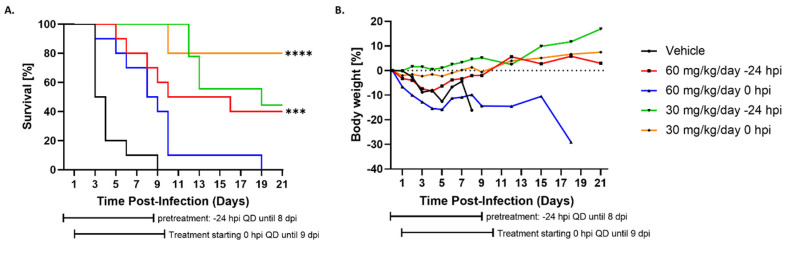
In vivo antiviral efficacy of Tilorone at different doses against RVFV infection in BALB/c mice. Mice (n = 10/group) were infected with 100 PFU RVFV ZH501 via the intraperitoneal (IP) route. Treatment with either 30 mg/kg/day or 60 mg/kg/day of Tilorone was initiated either 24 h before infection or immediately after infection. Tilorone or vehicle solution was administered once daily via the IP route for 9 days after initiation of treatment. (**A**) Survival of animals receiving Tilorone or vehicle. (**B**) Percent weight change of animals receiving Tilorone or vehicle. (**C**) Terminal viremia for euthanized moribund animals receiving Tilorone or vehicle. * *p* < 0.05, *** *p* < 0.001, **** *p* < 0.0001.

**Figure 4 microorganisms-10-00092-f004:**
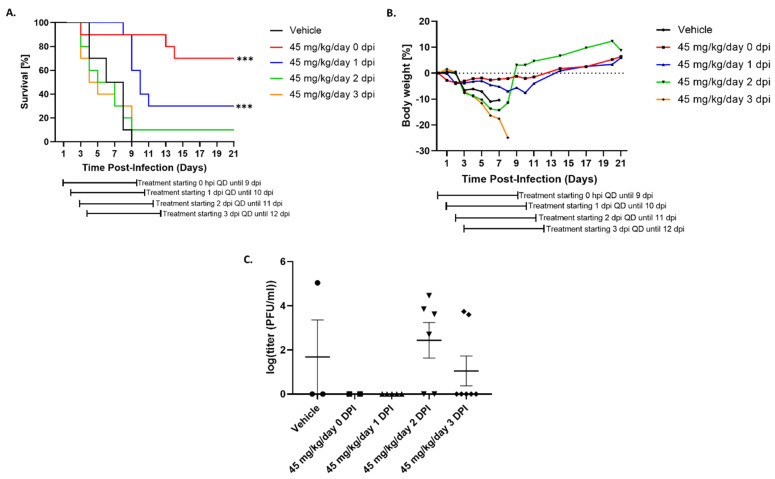
Therapeutic efficacy of Tilorone against RVFV infection in BALB/c mice. Mice (n = 10/group) were infected with 100 PFU RVFV ZH501 via the intraperitoneal (i.p.) route. Treatment of Tilorone was initiated immediately after infection, 1 DPI, 2 DPI, or 3 DPI. Tilorone at 45 mg/kg/day or vehicle was administered once daily via the IP route for 8 consecutive days after initiation of treatment. (**A**) Survival of animals receiving Tilorone or vehicle. (**B**) Percent weight change of animals receiving Tilorone or vehicle. (**C**) Terminal viremia for euthanized moribund animals receiving Tilorone or vehicle. *** *p* < 0.001.

**Figure 5 microorganisms-10-00092-f005:**
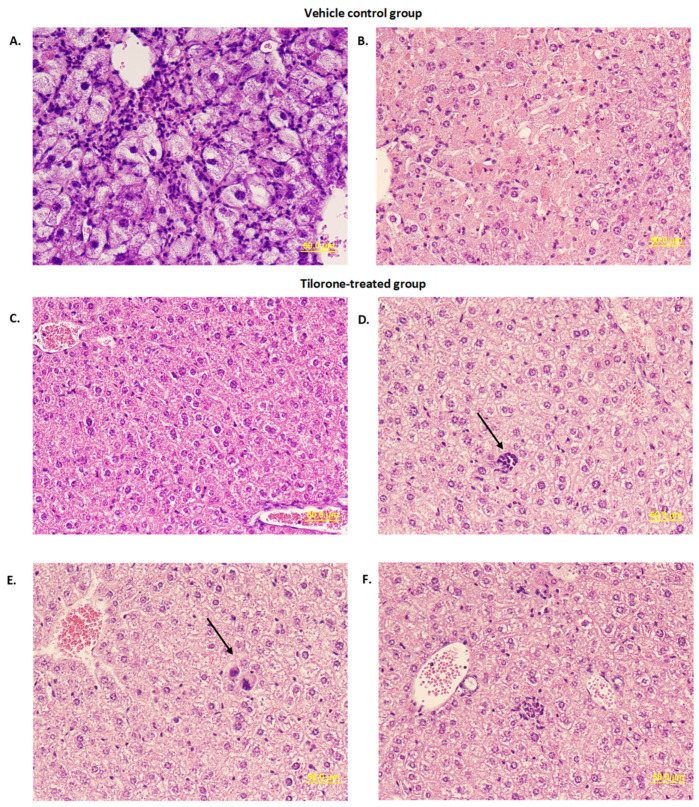
Histopathological changes reduced in Tilorone-treated mice. Formalin fixed tissues were embedded in paraffin and processed for H&E staining. Images represent livers of vehicle control and tilorone-treated mice. (**A**) Ballooned hepatocytes in vehicle-treated control mice at late stage. (**B**) Hepatocyte necrosis in vehicle-treated control mice. (**C**) Normal liver from Tilorone-treated survivor. (**D**) Erythroid cells in liver from Tilorone-treated survivor. (**E**) Megakaryocyte in liver from Tilorone-treated survivor. (**F**) Neutrophil infiltrate in liver from Tilorone-treated survivor.

**Table 1 microorganisms-10-00092-t001:** Tilorone-induced cytotoxicity in cell culture.

Cell Line	Time [HPI]	CC_50_ [μM]
**Vero CCL81**	24	>100
	48	30.76
	72	34.86
**A549**	24	>100
	48	11.23
	72	6.64

**Table 2 microorganisms-10-00092-t002:** In vitro antiviral efficacy of Tilorone against RVFV.

RVFV Strain	Cell Line	EC_50_ (μM)	EC_90_ (μM)	SI ^1^
**MP12**	Vero CCL81	0.67	3.08	>149
	A549	1.41	8.87	>71
**ZH501**	Vero CCL81	6.45	17.78	>16
	A549	6.31	31.62	>16

^1^: SI value is defined as SI = CC_50_/EC_50_; CC_50_ value listed in Table 1.

## Data Availability

No further datasets are available.

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
