# Peer review of "Tilorone-Dihydrochloride Protects against Rift Valley Fever Virus Infection and Disease in the Mouse Model"

_microorganisms, 2021, doi:10.3390/microorganisms10010092_

Round 1

Reviewer 1 Report

Johnson et al studied the feasibility of using Tilorone-dihydrochloride as an antiviral therapy against RVFV. In the paper they showed that Tilorone has an effect on the infectivity of the virus however this study lack some controls, such as Tilorone only animal group, which would provide good information to the present study. Few points are described below which must be clarified.

1. Regarding the toxicity of the compound in vitro

In figure 2 authors show the effect of Tilorone on cells and they state that they use 20 uM because that is the non-toxic concentration but that concentration is still toxic for A594 cells (Table 1) therefore the conclusions obtained from Figure 2.E, Tilorone being more effective on A594 than Vero cells, must be taken carefully since the effect observed in the viral titers at 48 and 72 hours might be due to the toxic effect of the compound on the viability of cells.

2. In vivo experiment with different concentrations

I think the experiment lacks an animal control group only inoculated with Tilorone to check the effect in vivo and to be able to verify what part of the mortality is due to this.

Do the authors have an explanation for the higher mortalitiy in the group inoculated with 60 mg at 0 hpi compared to the group with 60 mg at -24h? I assume that if it is an accumulative effect the -24 h should have a higher mortality as it happens in the 30 mg -24h group.

I think authors should include a graph with the viremia throughout all of the experiment or at least at some time points. I think this graph would provide a lot of useful information about the protection induced by Tilorone. Showing only data from dead animals reduces the possibility of knowing at what level it affects viral replication.

Also, what is the authors explanation for the animals that succumb to the inoculation with the virus but then didn’t present viremia both at early and late times after inoculation?

Why didn’t the authors study organs from these experiment? It would’ve been interesting to check if there were histopathological differences between groups with different doses.

3. In vivo experiment with 45 mg

Again, I would like to see data from the viremia of all the experiment. (Line 327)

I think a more detailed histopathological analysis would be very valuable. Analyzing more organs and not only brain and liver would provide information about toxic effects. Why didn’t the authors study organs from these experiment?

4. Discussion

In line 393 when authors talk about the different EC50 displayed against ZH501 and MP-12 they remark the differences showed for ZH501 while the difference against MP-12 is way higher. How do the authors explain the different EC50 for MP-12 in vero and A594 cells?

“Additionally, no viral RNA was detected in tissues for any survivors and no replicating virus was detected in serum samples from groups with significant survival across both in vivo studies.” At what times were serum samples taken? Why do the authors think no viremia was detected in any animal except for those in the control group even though animals died after inoculation with the virus?

If a toxic effect had been observed in the experiment with 60 mg and 30 mg administer at 0 hpi, why did the authors try 45 mg in the next experiment? Do they think the 10% reduction of survival is due to the increment of the dose?

Author Response

Johnson et al studied the feasibility of using Tilorone-dihydrochloride as an antiviral therapy against RVFV. In the paper they showed that Tilorone has an effect on the infectivity of the virus however this study lack some controls, such as Tilorone only animal group, which would provide good information to the present study. Few points are described below which must be clarified.

1. Regarding the toxicity of the compound in vitro

In figure 2 authors show the effect of Tilorone on cells and they state that they use 20 uM because that is the non-toxic concentration but that concentration is still toxic for A594 cells (Table 1) therefore the conclusions obtained from Figure 2.E, Tilorone being more effective on A594 than Vero cells, must be taken carefully since the effect observed in the viral titers at 48 and 72 hours might be due to the toxic effect of the compound on the viability of cells.

Re: We have simplified Figure 2 (see first comment from reviewer #2) and limit the described data for the first 24 HPI at which the CC50 is >100uM. Future studies will need to address the cumulative effect of cytotoxicity and its effect on RVFV replication.

2. In vivo experiment with different concentrations

I think the experiment lacks an animal control group only inoculated with Tilorone to check the effect in vivo and to be able to verify what part of the mortality is due to this.

Re: We agree with the reviewer that a Tilorone-only group would have been important to monitor side effects. We have previously performed a maximum tolerated dose (MTD) study in BALB/c mice, in which we found that the MTD is 100 mg/kg (Ekins et al 2018). In this study we observed that mice dosed with either 10, 50, or 100 mg/kg Tilorone survived, but demonstrated some adverse events, such as ruffled fur and hunched posture. Based on these data, we decided to not include a Tilorone-only group. This is information was included in the revised manuscript (lines 267 to 270) and we hope that it clarifies that Tilorone might contribute to certain toxic side effects, but that it should not contribute to mortality in our study at the concentrations tested.

Do the authors have an explanation for the higher mortalitiy in the group inoculated with 60 mg at 0 hpi compared to the group with 60 mg at -24h? I assume that if it is an accumulative effect the -24 h should have a higher mortality as it happens in the 30 mg -24h group.

Re: At the moment, we do not have any specific explanation for the observed differences in mortality between the 30 and 60 mg/kg doses and the different treatment times. However, similar observations were made in the study described by Ekins et al (2018), in which differences between treatment conditions were noted that were counterintuitive. Once we have a better understanding of the exact mechanism of action of Tilorone against RVFV infection, we might be able to better explain this observation. 

I think authors should include a graph with the viremia throughout all of the experiment or at least at some time points. I think this graph would provide a lot of useful information about the protection induced by Tilorone. Showing only data from dead animals reduces the possibility of knowing at what level it affects viral replication.

Also, what is the authors explanation for the animals that succumb to the inoculation with the virus but then didn’t present viremia both at early and late times after inoculation?

Re: We agree with the reviewers’ comment that monitoring the effect of Tilorone on RVFV viremia would be important. Unfortunately, these samples do not currently exist to address this concern and will need to be addressed in a future study performing serial sampling at different time points post infection. Regarding the lack of detectable viremia in mice that succumbed to infection, RVFV is causing a biphasic disease, targeting the liver first and which is typically the cause of death in non-protected animals. During this phase, viremia is detectable. However, mice which survive the hepatic course of the disease, often succumb to infection by developing neurological complications. During this phase of the disease, viremia is often not detectable. A statement in regards to both comments was added to the discussion (lines 479 to 490).

Why didn’t the authors study organs from these experiment? It would’ve been interesting to check if there were histopathological differences between groups with different doses.

Re: We agree with the reviewers’ comment that monitoring the effect of Tilorone on RVFV infection in tissues would be important. Unfortunately, these samples do not currently exist to address this concern and will need to be addressed in a future study performing serial sampling at different time points post infection.

3. In vivo experiment with 45 mg

Again, I would like to see data from the viremia of all the experiment. (Line 327)

I think a more detailed histopathological analysis would be very valuable. Analyzing more organs and not only brain and liver would provide information about toxic effects. Why didn’t the authors study organs from these experiment?

Re: Again, we agree with the reviewer that a comprehensive analysis of the effect of Tilorone of histopathological changes and on viremia would be important. However, the study design for the data presented in this manuscript was to determine if an antiviral effect of Tilorone can be detected in the mouse model. Future studies will need to be performed to generate those data.

4. Discussion

In line 393 when authors talk about the different EC50 displayed against ZH501 and MP-12 they remark the differences showed for ZH501 while the difference against MP-12 is way higher. How do the authors explain the different EC50 for MP-12 in vero and A594 cells?

Re: We thank the reviewer for raising this question. Previous studies have found that the effect of Tilorone to vary greatly depending on virus and cell type, with which our results are consistent. As the mechanism of action is currently unconfirmed, this likely contributes to the extensive variation of efficacies. This is described in lines 406 to 432.

“Additionally, no viral RNA was detected in tissues for any survivors and no replicating virus was detected in serum samples from groups with significant survival across both in vivo studies.” At what times were serum samples taken? Why do the authors think no viremia was detected in any animal except for those in the control group even though animals died after inoculation with the virus? 

Re: Please see our earlier response. A statement was added to the discussion (lines 479 to 490).

If a toxic effect had been observed in the experiment with 60 mg and 30 mg administer at 0 hpi, why did the authors try 45 mg in the next experiment? Do they think the 10% reduction of survival is due to the increment of the dose?

Re: We based the doses of Tilorone tested in this study on data that were generated in Ekins et al (2018). After the initial study with 60 mg/kg and 30 mg/kg, we decided to test at a concentration mid-way between the first two. While the 30 mg/kg group given at 0HPI resulted in promising survival data, we thought that a slightly higher dose might increase the chance of higher survival, specifically if we initiate dosing at different times post infection. Future studies might need to be performed to determine the dose-response of Tilorone in RVFV-infected mice.

Reviewer 2 Report

The manuscript of Johnson KN et al reports the in vitro and in vivo antiviral activity of Tilorone against the Rift Valley Fever virus (RVFV) showing a promising efficacy for this potentially broad-range drug. The paper is of interest for the developing of therapeutics against emerging viral infections.

Although the manuscript is well written and results might support Authors’ conclusions, some sections require a revision.

Authors shown that Tilorone inhibits RVFV in vitro both in interferon competent and interferon defective cell lines.  Treatment is associated to a relevant cytotoxicity (as reported in table 1) after 48 and 72 hours of cell exposition to Tilorone. In this context, data reported in figure 2 about the viral titer measured at 72 HPI are difficult to be evaluated due to the cumulative effect of the cytotoxicity. Authors should limit the analyses to 24h after the drug treatment adding a qt-PCR on viral RNA inside cells to evaluate the inhibition of RVFV replication, In particular when the virus progeny yield is close to the detection limit of the plaque assay. I addition, the selective index of Tilorone (against the MP12 strain) reported in lines 182 and 183 for Vero CCL81 and A459 cells is different to the one reported in table 2. Probably results for the MP12 might be deleted, they are not essential for the in vitro and in vivo data with the pathogenic strains.

Is there any evidence of an additive effect of compound cytotoxicity and cytopathogenicity of infection on cell survival? It could also be involved in the absence of survivors with the 0 HPI co-treatment group (at 60 mg/kg/day)

Please, add in math & meth (paragraph 2.5) the volume of virus inoculum.

Authors suggest that the optimization of the dosing regimen could improve the results on the mice survival. Considering that Tilorone is used as antivirals in some Countries, are available data on the serum concentrations reach in humans? What is the usual route of administration in humans?

In the paragraph of “histopathological evaluation” and in the Conclusions, no comments are available about the tissue alterations detected in Tilorone-treated mice that died. Is there any difference in comparison to vehicle-treated group?

Author Response

The manuscript of Johnson KN et al reports the in vitro and in vivo antiviral activity of Tilorone against the Rift Valley Fever virus (RVFV) showing a promising efficacy for this potentially broad-range drug. The paper is of interest for the developing of therapeutics against emerging viral infections.

Although the manuscript is well written and results might support Authors’ conclusions, some sections require a revision.

Authors shown that Tilorone inhibits RVFV in vitro both in interferon competent and interferon defective cell lines.  Treatment is associated to a relevant cytotoxicity (as reported in table 1) after 48 and 72 hours of cell exposition to Tilorone. In this context, data reported in figure 2 about the viral titer measured at 72 HPI are difficult to be evaluated due to the cumulative effect of the cytotoxicity. Authors should limit the analyses to 24h after the drug treatment adding a qt-PCR on viral RNA inside cells to evaluate the inhibition of RVFV replication, In particular when the virus progeny yield is close to the detection limit of the plaque assay. I addition, the selective index of Tilorone (against the MP12 strain) reported in lines 182 and 183 for Vero CCL81 and A459 cells is different to the one reported in table 2. Probably results for the MP12 might be deleted, they are not essential for the in vitro and in vivo data with the pathogenic strains.

Re: We thank the reviewer for the suggestion to limit the data in Figure 2. Accordingly, we simplified Figure 2 and only show data for the initial 24 HPI. Future studies will need to be performed to address the cumulative effect of cytotoxicity and its effect on RVFV replication. We also agree with the reviewer that analysis by PCR would be helpful to monitor the effect of Tilorone on RVFV infection. These studies are planned for future evaluation of the mechanism of action of Tilorone on RVFV infection, as well as further characterization of the antiviral effect in vivo by performing serial sampling studies to evaluate the effect of RVFV viremia and histopathological changes in tissues. We also thank the reviewer for catching the error in reported SI values. The values in Table 2 are correct and we corrected the information in lines 184 and 185.

Is there any evidence of an additive effect of compound cytotoxicity and cytopathogenicity of infection on cell survival? It could also be involved in the absence of survivors with the 0 HPI co-treatment group (at 60 mg/kg/day)

Re: Currently, we do not have any evidence to support or dispute an additive effect. Future studies focusing on mechanism of action will need to address this question.

Please, add in math & meth (paragraph 2.5) the volume of virus inoculum.

Re: The missing information has been added (line 127).

Authors suggest that the optimization of the dosing regimen could improve the results on the mice survival. Considering that Tilorone is used as antivirals in some Countries, are available data on the serum concentrations reach in humans? What is the usual route of administration in humans?

Re: We added a brief statement indicating that Tilorone is administered orally (lines 504 to 508). We were not able to find readily available data on serum concentrations or even safety or efficacy data in humans. Most of the literature is in Russian, and need to be translated first.

In the paragraph of “histopathological evaluation” and in the Conclusions, no comments are available about the tissue alterations detected in Tilorone-treated mice that died. Is there any difference in comparison to vehicle-treated group?

Re: We thank the reviewer for this comment. From the limited number of tissues that were collected during the efficacy studies, we did not detect any differences other than what is described in the Results. Future studies are planned to perform serial sampling of tissues during the acute phase of the disease to allow a better comparison between treated and non-treated mice, as well as to include IHC staining.

Round 2

Reviewer 2 Report

Authors corrected the manuscript as requested and provided data/info to answer to my concerns.